

# Aerosol retrieval over snow using RemoTAP

Zihan Zhang[1,2], Guangliang Fu[1], Otto Hasekamp[1]

[1]SRON Netherlands Institute for Space Research, Leiden, 2333CA, the Netherlands
[2]School of Earth and Space Science, Peking University, Beijing, 100871, China

*Correspondence to*: Otto Hasekamp (o.p.hasekamp@sron.nl)

**Abstract.** In order to conduct accurate aerosol retrieval over snow, the Remote Sensing of Trace Gases and Aerosol Products (RemoTAP) algorithm developed by SRON Netherlands Institute for Space Research is extended with a Bi-directional Reflection Distribution Function (BRDF) for snow surfaces. The capability of the extended algorithm is validated with both synthetic measurements and real satellite measurements from PARASOL and a comparison has been made to retrievals with

the baseline RemoTAP (without snow kernel). For retrievals on real PARASOL observations, we use pixels over AERONET stations for validation and we use the MODIS snow cover products to identify pixels over snow. We evaluate the retrieved aerosol optical thickness (AOT) at 550 nm ($\tau_{550}$), single-scattering albedo (SSA) at 550 nm ($\omega_{550}$), and Angstrom exponent (AE) for 440 nm – 870 nm ($AE_{440-870}$). Both the experiments with synthetic- and real data show that the extended RemoTAP maintains capability on snow-free pixels and has obvious advantages on accuracy and fraction of successful retrievals for

retrieval over snow, especially over surfaces with snow cover > 75%. According to the real data experiment, we find that the retrieval algorithm has difficulty in fitting the PARASOL 1020 nm band, where snow reflectance is significantly lower than that for the visible bands. When we perform a 4-band retrieval (490 nm, 565 nm, 670 nm, 865 nm) with the extended RemoTAP, we obtain a good retrieval result for $\tau_{550}$, $\omega_{550}$ and $AE_{440-870}$. Therefore, the 4-band retrieval with the extended RemoTAP is recommended for aerosol retrieval over snow.

## 1 Introduction

Global climate change is greatly influenced by aerosol through aerosol cloud interaction (e.g., Twomey, 1974; Li et al., 2011; Rosenfeld et al., 2014; Hasekamp et al., 2019a; Gryspeerdt et al., 2020; Quaas et al., 2020) and aerosol radiation interaction (e.g., Koren et al., 2004; Yu et al., 2006; Myhre, 2009; Guo et al., 2017; Lacagnina et al., 2017; Witthuhn et al., 2021). These effects cause a significant radiative forcing of climate (Bellouin et al., 2020; Haywood et al., 2021). According to the sixth

Assessment Report (AR6) of Intergovernmental Panel on Climate Change (IPCC), aerosols still represent the largest uncertainty in our quantification of global climate change (IPCC, 2023). Therefore, in order to better study climate change, it is essential to give an accurate estimate of global aerosol properties. Multi-Angle Polarimetric (MAP) satellite measurements provide the richest set of information on aerosol properties from a passive remote sensing point of view. So far, the only MAP instrument that has provided a multi-year dataset has been POLDER-3/PARASOL (here-after referred to as PARASOL),

which was active from 2005-2013. In the near future a number of new MAP instruments will be launched, including SPEXone



(Hasekamp et al., 2019b) and HARP-2 (Mcbride et al., 2020) on the NASA PACE mission (Werdell et al., 2019) and the 3MI instrument (Fougnie et al., 2018) on METOP-SG.

After years of development, there exist a number of aerosol retrieval algorithms which are available for multi-angle and multi-spectral polarization sensors. The algorithms can be classified into two groups: algorithms based on lookup-tables (LUTs)
and full inversion algorithms. LUT-based algorithms (Herman et al., 1997; Deuzé et al., 2000; Waquet et al., 2016) provide faster calculation, but are less accurate than full inversion algorithms. Among all the full inversion algorithms are the Microphysical Aerosol Properties from Polarimetry (MAPP) algorithm (Stamnes et al., 2018), the algorithm developed at the Jet Propulsion Laboratory (JPL) (Xu et al., 2016; Xu et al., 2017; Xu et al., 2018), the Generalized Retrieval of Aerosol and Surface Properties (GRASP) algorithm (Dubovik et al., 2011; Dubovik et al., 2014; Chen et al., 2020; Dubovik et al., 2021)
and the Remote-sensing of Trace-gas and Aerosol Product (RemoTAP) algorithm developed by SRON (Hasekamp et al., 2011; Fu and Hasekamp, 2018; Fu et al., 2020; Lu et al., 2022). Till now, only the RemoTAP algorithm and the GRASP algorithm have demonstrated capability at the global scale.

In order to generate global aerosol products, aerosol retrieval over snow remains an important challenge for the above-mentioned algorithms. Until now, studies on aerosol retrieval over snow have focused mainly on intensity sensors (Istomina
et al., 2011; Mei et al., 2013), focusing on AOT retrievals in the arctic region.

In our paper, we extend the RemoTAP algorithm to carry out aerosol retrieval over snow from MAP measurements of PARASOL. We evaluate the capability of the extended RemoTAP algorithm using synthetic observations as well as real PARASOL retrievals which are validated by AERONET. The paper is organized as follows: Section 2 introduces the methodology of RemoTAP for aerosol retrieval over snow; Section 3 describes the satellite data and ancillary data used in our
real data retrieval and the validation data; Section 4 shows results for synthetic retrievals; Section 5 shows the results for real PARASOL retrievals and provides a recommended routine for aerosol retrieval over snow from PARASOL; Section 6 discusses the results of the paper and future research, and concludes the paper.

## 2 Methodology

### 2.1 Forward model

For aerosol retrieval over snow, the detailed information for the extended RemoTAP algorithm is described below.

The aim of the RemoTAP algorithm is to retrieve a state vector $\boldsymbol{x}$ from measurement vector $\boldsymbol{y}$ by inverting the forward model $\mathbf{F}$:

$$\boldsymbol{y} = \mathbf{F}(\boldsymbol{x}) + \boldsymbol{e_y}, \tag{1}$$

where $\boldsymbol{e_y}$ refers to the measurement error vector.

The measurement by polarization sensors is described by the intensity vector $[I, Q, U, V]$ (Stokes, 1851). For Earth
observation, $V$ parameter can be ignored in most cases. In this study, the Stokes parameters are calculated by the SRON





radiative transfer model LINTRAN v2.0 (Schepers et al., 2014). The measurement vector $\boldsymbol{y}$ contains parameters including the Top-of-Atmosphere (ToA) apparent reflectance $\rho$ and Degree of Linear Polarization (DoLP) at different spectral bands and different observation geometries:

$$\rho = \frac{I}{E_0}, \tag{2}$$

$$DoLP = \frac{\sqrt{Q^2 + U^2}}{I}, \tag{3}$$

where, $E_0$ is the ToA solar irradiance perpendicular to the solar beam.

The state vector $\boldsymbol{x}$ contains parameters related to aerosol and surface characteristics. To describe aerosol properties in the state vector, following Lu et al., (2022), 3 log-normal modes are applied: (1) Mode 1 is a fine mode for which the state vector includes aerosol column number ($N_{\mathrm{aer}}$), effective radius ($r_{\mathrm{eff}}$), effective variance ($v_{\mathrm{eff}}$), spherical fraction ($f_{\mathrm{sph}}$), aerosol layer height ($z_{\mathrm{aer}}$) and the refractive index coefficients corresponding to the standard spectra (D'almeida et al., 1991; Kirchstetter et al., 2004) of Inorganic aerosol, Black Carbon (imaginary part) and Organic Carbon (imaginary part). (2) Mode 2 is an insoluble

coarse mode, which consists of non-spherical dust. The state vector includes $N_{\mathrm{aer}}$, $r_{\mathrm{eff}}$, $v_{\mathrm{eff}}$, $z_{\mathrm{aer}}$ and refractive index coefficients corresponding to the standard spectra of Dust (imaginary part). The fixed parameter is $f_{\mathrm{sph}} = 0$ and the coefficient for the real part of Dust refractive index is fixed to 1. $z_{\mathrm{aer}}$ is assumed to be the same for mode 1 and mode 2. (3) Mode 3 is a soluble coarse mode. The state vector includes $N_{\mathrm{aer}}$, $r_{\mathrm{eff}}$, $v_{\mathrm{eff}}$ and the refractive index coefficient corresponds to the standard spectra of Inorganic aerosol. The fixed parameters are $f_{\mathrm{sph}} = 1$ and $z_{\mathrm{aer}} = 500$ m.

To describe the surface, the surface reflection matrix $\mathbf{R}_{\mathrm{surf}}(\lambda, \theta_{\mathrm{s}}, \theta_{\mathrm{v}}, \varphi)$ is given by:

$$\mathbf{R}_{\mathrm{surf}}(\lambda, \theta_{\mathrm{s}}, \theta_{\mathrm{v}}, \varphi) = r_{11}(\lambda, \theta_{\mathrm{s}}, \theta_{\mathrm{v}}, \varphi)\mathbf{D} + \mathbf{R}_{\mathrm{pol}}(\theta_{\mathrm{s}}, \theta_{\mathrm{v}}, \varphi), \tag{4}$$

where $\lambda$ is the wavelength, $\theta_{\mathrm{s}}$ and $\theta_{\mathrm{v}}$ are the sun zenith angle and view zenith angle respectively, $\varphi$ is the relative azimuth angle, $\mathbf{D}$ is a null matrix except $\mathbf{D}_{11} = 1$, $r_{11}(\lambda, \theta_{\mathrm{s}}, \theta_{\mathrm{v}}, \varphi)$ is described by the Ross-Li BRDF model, extended by a snow kernel:

$$r_{11}(\lambda, \theta_{\mathrm{s}}, \theta_{\mathrm{v}}, \varphi) = A(\lambda)\big[1 + k_{\mathrm{geo}}f_{\mathrm{geo}}(\theta_{\mathrm{s}}, \theta_{\mathrm{v}}, \varphi) + k_{\mathrm{vol}}f_{\mathrm{vol}}(\theta_{\mathrm{s}}, \theta_{\mathrm{v}}, \varphi) + k_{\mathrm{snow}}f_{\mathrm{snow}}(\theta_{\mathrm{s}}, \theta_{\mathrm{v}}, \varphi)\big], \tag{5}$$

where $A(\lambda)$ is the isotropic reflectance, $f_{\mathrm{geo}}(\theta_{\mathrm{s}}, \theta_{\mathrm{v}}, \varphi)$ and $f_{\mathrm{vol}}(\theta_{\mathrm{s}}, \theta_{\mathrm{v}}, \varphi)$ are respectively the geometric (Li-Sparse) kernel and volumetric (Ross-Thick) kernel function (Wanner et al., 1995), $f_{\mathrm{snow}}(\theta_{\mathrm{s}}, \theta_{\mathrm{v}}, \varphi)$ is the snow kernel function (Jiao et al., 2019),

$k_{\mathrm{geo}}, k_{\mathrm{vol}}, k_{\mathrm{snow}}$ are the coefficients for the Li-Sparse, Ross-Thick and snow kernel respectively. It is important to note that the inclusion of $k_{\mathrm{snow}}f_{\mathrm{snow}}(\theta_{\mathrm{s}}, \theta_{\mathrm{v}}, \varphi)$ is the major difference between the baseline RemoTAP and the extended RemoTAP of the present work. The Li-Sparse kernel function $f_{\mathrm{geo}}(\theta_{\mathrm{s}}, \theta_{\mathrm{v}}, \varphi)$, Ross-Thick kernel function $f_{\mathrm{vol}}(\theta_{\mathrm{s}}, \theta_{\mathrm{v}}, \varphi)$ and snow kernel function $f_{\mathrm{snow}}(\theta_{\mathrm{s}}, \theta_{\mathrm{v}}, \varphi)$ are given in Eq. (6) and Eq. (7):

$$f_{\mathrm{geo}}(\theta_{\mathrm{s}}, \theta_{\mathrm{v}}, \varphi) = O(\theta_{\mathrm{s}}', \theta_{\mathrm{v}}', \varphi) - \left[\sec\theta_{\mathrm{s}}' + \sec\theta_{\mathrm{v}}' - \frac{1}{2}(1 - \cos\Theta')\sec\theta_{\mathrm{s}}'\sec\theta_{\mathrm{v}}'\right], \tag{6}$$

$$f_{\mathrm{vol}}(\theta_{\mathrm{s}}, \theta_{\mathrm{v}}, \varphi) = \frac{(\Theta - \pi/2)\cos(\pi - \Theta) + \sin(\pi - \Theta)}{\cos\theta_{\mathrm{s}} + \cos\theta_{\mathrm{v}}} - \frac{\pi}{4}, \tag{7}$$





where $\Theta$ is the scattering angle ($\cos\Theta = -\cos\theta_s\cos\theta_v - \sin\theta_s\sin\theta_v\cos\varphi$), $\theta_s{'}, \theta_v{'}$ are the equivalent zenith angles (the

transformation is $\theta' = \tan^{-1}(b/r \cdot \tan\theta)$ for sun zenith angle and view zenith angle respectively, $b$ and $r$ are vertical and horizontal crown radius respectively), $O(\theta_s{'}, \theta_v{'}, \varphi)$ is the overlap function given by Li and Strahler (1992).

The snow kernel function $f_{\text{snow}}(\theta_s, \theta_v, \varphi)$ is given by Jiao et al. (2019) in Eq. (8):

$$f_{\text{snow}}(\theta_s, \theta_v, \varphi) = R_0(\theta_s, \theta_v, \varphi)\left[1 - \alpha \cdot \cos(\pi - \Theta) \cdot e^{-\cos(\pi - \Theta)}\right] + 0.4076 \cdot \alpha - 1.1081, \tag{8}$$

where $R_0(\theta_s, \theta_v, \varphi)$ is the reflectance for a semi-infinite, non-absorbing snow layer at zero absorption and $\alpha$ is an empirical parameter used to correct $R_0(\theta_s, \theta_v, \varphi)$ for the underestimation of the reflectance in the forward-scattering direction.

$R_0(\theta_s, \theta_v, \varphi)$ is given by Kokhanovsky et al. (2005):

$$R_0(\theta_s, \theta_v, \varphi) = \frac{K_1 + K_2 \cdot (\cos\theta_s + \cos\theta_v) + K_3 \cdot \cos\theta_s \cdot \cos\theta_v + P(\Theta)}{4(\cos\theta_s + \cos\theta_v)}, \tag{9}$$

$$P(\Theta) = 11.1 \cdot e^{-0.087 \cdot \Theta} + 1.1 \cdot e^{-0.014 \cdot \Theta}, \tag{10}$$

where $K_1$, $K_2$ and $K_3$ are three constants. In our algorithm, $\alpha = 0.3$, $K_1 = 1.247$, $K_2 = 1.186$ and $K_3 = 5.157$ are fixed when calculating snow kernel, as suggested by Jiao et al. (2019) and Kokhanovsky et al. (2005).

$\mathbf{R}_{\text{pol}}(\theta_s, \theta_v, \varphi)$ in Eq. (4) is given by:

$$\mathbf{R}_{\text{pol}}(\theta_s, \theta_v, \varphi) = B_{\text{pol}}\frac{\exp\left[-\tan\left(\frac{\pi - \Theta}{2}\right)\right]\exp(-v)\mathbf{F}_p(m, \Theta)}{4(\cos\theta_s + \cos\theta_v)}, \tag{11}$$

where $B_{\text{pol}}$ is the free linear parameter, $\mathbf{F}_p(m, \Theta)$ is the Fresnel scattering matrix and $m$ is the refractive index (Maignan et al.,

2009). In our experiment, $m = 1.5$ and $v = 0.1$ are fixed when calculating $\mathbf{R}_{\text{pol}}$.

To characterize the surface properties, we include $A(\lambda), k_{\text{geo}}, k_{\text{vol}}, k_{\text{snow}}$ and $B_{\text{pol}}$ as the surface parameters in the state vector. Theoretically, the surface model still maintains the ability of depicting snow-free surfaces, because $k_{\text{snow}}$ is fitted in the retrieval (for snow-free surfaces, it should retrieve $k_{\text{snow}} = 0$).

The aerosol and surface parameters in the state vector $\boldsymbol{x}$ are shown in Table 1.



**Table 1: Parameters in the state vector $x$ utilized in the retrieval. $c_{1,\dots}$ correspond to coefficients for standard refractive index of Inorganic aerosol (INORG), Black Carbon (BC), Organic Carbon (OC), Dust (DU).**

| | Property | | Full name | A-priori |
|---|---|---|---|---|
| Retrieved aerosol properties | Fine mode (mode 1) | $N_{aer}$ | Aerosol column number (mode 1) | LUT-retrieval |
| | | $r_{eff}$ | Effective radius (mode 1) | 0.1 |
| | | $v_{eff}$ | Effective variance (mode 1) | 0.2 |
| | | $f_{sph}$ | Spherical fraction (mode 1) | 0.95 |
| | | $z_{aer}$ | Aerosol layer height (mode 1) | 2000 |
| | | $c_1$ (INORG real) | Refractive index coefficient (INORG real) | 0.95 |
| | | $c_2$ (BC imaginary) | Refractive index coefficient (BC imaginary) | 0.05 |
| | | $c_3$ (OC imaginary) | Refractive index coefficient (OC imaginary) | 0.1 |
| | Coarse insoluble mode (mode 2) | $N_{aer}$ | Aerosol column number (mode 2) | LUT-retrieval |
| | | $r_{eff}$ | Effective radius (mode 2) | 1.5 |
| | | $v_{eff}$ | Effective variance (mode 2) | 0.6 |
| | | $z_{aer}$ | Aerosol layer height (mode 2) | 2000 |
| | | $c_1$ (DU imaginary) | Refractive index coefficient (DU imaginary) | 0.1 |
| | Coarse soluble mode (mode 3) | $N_{aer}$ | Aerosol column number (mode 3) | LUT-retrieval |
| | | $r_{eff}$ | Effective radius (mode 3) | 3.0 |
| | | $v_{eff}$ | Effective variance (mode 3) | 0.6 |
| | | $c_1$ (INORG) | Refractive index coefficient (INORG) | 1.0 |
| Retrieved surface properties | $k_{geo}$ | | Geometric kernel coefficient | 0.2 |
| | $k_{vol}$ | | Volumetric kernel coefficient | 0.5 |
| | $k_{snow}$ | | Snow kernel coefficient | 0.9 |
| | $A(\lambda)$ | | Isotropic reflectance | $\begin{cases} 0.9 & (\lambda < 800) \\ 0.6 & (\lambda \geq 800) \end{cases}$ |
| | $B_{pol}$ | | Free linear parameter for BPDF | 2.0 |

## 2.2 Inversion algorithm

To retrieve the state vector $x$ from the measurement $y$, a damped Gauss-Newton iteration method with Phillips-Tikhonov regularization is employed (Hasekamp et al., 2011; Fu and Hasekamp, 2018). We shortly summarize the method here. The aim of the inversion algorithm is to find the solution $\hat{x}$ of the minimization-optimization problem:

$$\hat{x} = \min_x \left[ \left\| S_y^{-\frac{1}{2}}(F(x) - y) \right\|^2 + \gamma^2 \left\| W^{-\frac{1}{2}}(x - x_a) \right\|^2 \right], \tag{12}$$





where $\boldsymbol{x}_a$ is the a-priori state vector, $\mathbf{W}$ is the diagonal weight matrix in order to remove the order-of-magnitude difference of each state parameters ($W_{ii} = 1/x_{a,i}$), $\mathbf{S}_y$ is the diagonal measurement error covariance matrix which is related to the sensor, $\gamma$ is the regularization parameter (Hasekamp et al., 2011).

Since the forward model $\mathbf{F}$ is nonlinear, an iterative strategy is utilized to conduct the inversion. For each iteration $n$, the linear approximation of the forward model is given in Eq. (13):

$$\mathbf{F}(\boldsymbol{x}) \approx \mathbf{F}(\boldsymbol{x}_n) + \mathbf{K}(\boldsymbol{x} - \boldsymbol{x}_n), \tag{13}$$

where $\mathbf{K}$ is the Jacobian matrix containing the derivatives of the forward model $\mathbf{F}$ with respect to the parameters in state vector $\boldsymbol{x}$, and $K_{ij} = \frac{\partial F_i}{\partial x_j}(\boldsymbol{x}_n)$.

With linear approximation, Eq. (12) can be simplified:

$$\tilde{\boldsymbol{x}}_{n+1} = \min_{\tilde{\boldsymbol{x}}}\left[\left\|\tilde{\mathbf{K}}(\tilde{\boldsymbol{x}} - \tilde{\boldsymbol{x}}_n) - \tilde{\boldsymbol{y}}\right\|^2 + \gamma^2\|\tilde{\boldsymbol{x}} - \tilde{\boldsymbol{x}}_a\|^2\right], \tag{14}$$

where $\tilde{\boldsymbol{x}} = \mathbf{W}^{-\frac{1}{2}}\boldsymbol{x}$, $\tilde{\boldsymbol{y}} = \mathbf{S}_y^{-\frac{1}{2}}(\boldsymbol{y} - \mathbf{F}(\boldsymbol{x}_n))$ and $\tilde{\mathbf{K}} = \mathbf{S}_y^{-\frac{1}{2}}\mathbf{K}\mathbf{W}^{\frac{1}{2}}$.

The solution to Eq. (14) for iteration step $n$ is given by:

$$\tilde{\boldsymbol{x}}_{n+1} = \tilde{\boldsymbol{x}}_n + \Lambda\left(\tilde{\mathbf{K}}^T\tilde{\mathbf{K}} + \gamma^2\mathbf{I}\right)^{-1}\left[\tilde{\mathbf{K}}^T\tilde{\boldsymbol{y}} - \gamma^2(\tilde{\boldsymbol{x}}_n - \tilde{\boldsymbol{x}}_a)\right], \tag{15}$$

where $\mathbf{I}$ is the identity matrix, $\Lambda$ is the filter factor ($0 \leq \Lambda \leq 1$), which is utilized to control the step size per iteration. For each iteration, the optimal $\Lambda$ and $\gamma$ are chosen via goodness-of-fit assessment by comparing $\chi^2$ given in Eq. (16):

$$\chi^2 = \frac{1}{N_{\text{meas}}}\sum_{i=1}^{N_{\text{meas}}}\left(\frac{y_i - F_i}{e_i}\right)^2, \tag{16}$$

where $N_{\text{meas}}$ is the length of measurement vector $\boldsymbol{y}$, $y_i$ and $F_i$ refer to the measurements and the results of forward model

respectively, $e_i$ is the measurement uncertainties determined by the instrument. After the final iteration step, we use $\chi^2 < 5$ as the threshold to determine whether the retrieval was successful or not. For a set of retrievals, the fraction of successful retrievals (FoSR) is determined based on this filter.

## 3 Data

### 3.1 PARASOL data

The microsatellite Polarization & Anisotropy of Reflectances for Atmospheric Sciences coupled with Observations from a Lidar (PARASOL), equipped with POLarization and Directionality of the Earth's Reflectances-3 instrument, was launched on 18[th] December 2004 (Fougnie et al., 2007; Lier and Bach, 2008). PARASOL includes 9 spectral bands and 3 bands contain polarization information (the details of the bands are shown in Table 2).





**Table 2: Spectral band information for PARASOL**

| PARASOL band (nm) | Central Wavelength (nm) | Band Width (nm) | Polarization |
|---|---|---|---|
| 443 | 443.9 | 13.5 | × |
| 490 | 491.5 | 16.5 | √ |
| 565 | 563.9 | 15.5 | × |
| 670 | 669.9 | 15.0 | √ |
| 763 | 762.8 | 11.0 | × |
| 765 | 762.5 | 38.0 | × |
| 865 | 863.4 | 33.5 | √ |
| 910 | 906.9 | 21.0 | × |
| 1020 | 1019.4 | 17.0 | × |


PARASOL provides multidirectional and multispectral data from December 2004 to December 2013, which has been used for aerosol retrieval (e.g., Tanré et al., 2011; Dubovik et al., 2011; Fu and Hasekamp, 2018; Chen et al., 2020), and the resolution is approximately 6 km in the nadir. In our experiments, we consider 5 bands (490 nm, 565 nm, 670 nm, 865 nm, 1020 nm), because 443 nm band suffers from stray light which may be in particular problematic over bright snow surfaces

(Fougnie et al., 2007), 763 nm and 765 nm are mainly used to retrieve cloud oxygen pressure and 910 nm is usually used to retrieve the water vapor (Leroy et al., 1997).

### 3.2 Ancillary data

As input to the retrieval, ancillary data are needed including surface pressure, and profiles of air temperature, relative humidity after moist, ozone mass mixing ratio. These data are obtained from the Modern-Era Retrospective analysis for Research and

Applications Version 2 (MERRA-2) (Gelaro et al., 2017). Additionally, the cloud fraction is from the MODerate-resolution Imaging Spectroradiometer (MODIS). In order to make a reliable cloud screening, only pixels with cloud fraction under 0.2 are used in our experiments.

### 3.3 Snow cover data

During the process of retrieval and validation, pixels with snow cover are selected based on the MODIS/Aqua Snow Cover

Daily L3 Global 500m SIN Grid (MYD10A1) product (Hall et al., 2019). Each tile is generated from MODIS/Aqua Snow Cover 5-Min L2 Swath 500m (MYD10_L2) product. Given the row and column in a MODIS tile, the 500 m MODIS sinusoidal grid is converted to PARASOL sinusoidal grid. It is important to note that the MODIS snow cover product has been pre-processed with a cloud filtering procedure.





### 3.4 AERONET data

In this paper, the main concern is aerosol retrieval over snow. The RemoTAP-retrieved aerosol properties are validated with AERONET data (Holben et al., 1998). AERONET provides 3 levels of data quality: level 1.0, level 1.5 and level 2.0. Level 1.0 provides unscreened data which have been rarely used for validation. Level 1.5 provides near-real-time, cloud-screened data with instrument quality control. Level 2.0 provides quality-assured data on the basis of level 1.5 by applying pre-field and post-field calibrations. The retrieved AOT and AE are validated with AERONET direct sun level 2.0 AOT data (Giles et al.,
2019). The retrieved SSA is validated with AERONET-Inversion level 2.0 SSA data (Sinyuk et al., 2020).

### 3.5 Data pre-processing

In the real data experiments, the PARASOL data are pre-processed with in steps. The first step is to match global PARASOL L1 data with global AERONET data. For each successfully-matched group of pixels (here-after referred to as a colocation), the difference between the measurement time of AERONET data and PARASOL pixels is within 1 hour and the distance
between AERONET data and PARASOL pixels is within 20 km. The second step is to match MODIS snow cover data with AERONET-colocated PARASOL data, thus divide the colocated PARASOL data into snow pixels and snow-free pixels.

### 4 Synthetic data experiments

The forward model of RemoTAP is used to generate the synthetic PARASOL measurement and noise is subsequently added according to a Gaussian distribution. For ToA reflectance, the simulated noise (1 standard deviation) is 1%, and for DoLP it
is 0.007 (absolute). The set of synthetic measurements contains 1000 pixels with randomly-generated input land properties, aerosol properties, auxiliary properties and geometry properties. The settings for the properties are shown in Table 3.

**Table 3: Observation geometry, aerosol properties and surface properties used to create synthetic PARASOL observations.**

| Property | Minimum | Maximum |
|---|---|---|
| $\theta_s$ | 10 | 70 |
| $\theta_v$ | / | / |
| $\varphi$ | / | / |
| $k_{geo}$ | / | / |
| $k_{vol}$ | / | / |
| $k_{snow}$ | -0.2 | 1.5 |
| $A(\lambda)$ | / | / |
| $\tau_{550}$ (mode 1) | 0.005 | 1.0 |
| $\tau_{550}$ (mode 2) | 0.0025 | 0.25 |
| $\tau_{550}$ (mode 3) | 0.0025 | 0.25 |





| | | |
|---|---|---|
| $r_{eff}$ (mode 1) | 0.1 | 0.3 |
| $r_{eff}$ (mode 2) | 0.8 | 1.5 |
| $r_{eff}$ (mode 3) | 1.5 | 4.0 |
| $v_{eff}$ (mode 1) | 0.1 | 0.3 |
| $v_{eff}$ (mode 2) | 0.6 | 0.6 |
| $v_{eff}$ (mode 3) | 0.6 | 0.6 |
| $f_{sph}$ (mode 1) | 1.0 | 1.0 |
| $f_{sph}$ (mode 2) | 0.0 | 0.0 |
| $f_{sph}$ (mode 3) | 1.0 | 1.0 |
| $z_{aer}$ (mode 1) | 1000 | 6000 |
| $z_{aer}$ (mode 2) | 1000 | 6000 |
| $z_{aer}$ (mode 3) | 500 | 500 |

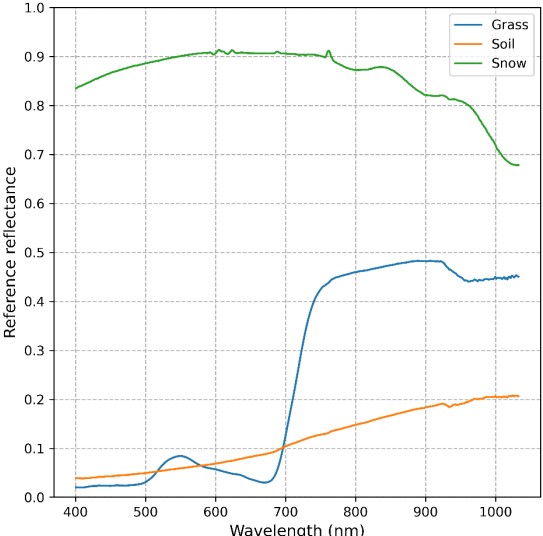

**Figure 1: Reference reflectance for vegetation, soil and snow. The snow data are downloaded from National Snow and Ice Data Center (NSIDC) (https://nsidc.org/data/hma_sbrf/versions/1) and the soil and vegetation data are downloaded from ASTER spectral library (https://speclib.jpl.nasa.gov/).**

The surface properties in the synthetic data set are created by mixing the contribution of the surface reflection by vegetation, soil, and snow. By controlling the fraction of vegetation, soil and snow, 4 sets of synthetic measurements are created, where the detailed information for these 4 synthetic measurements are listed in Table 4. The isotropic part of the BRDF for snow, soil, and vegetation, as used in this study, is shown in Fig. 1. For the directional BRDF parameters for soil and vegetation, we use the values found by Litvinov et al. (2011).



**Table 4: Descriptions for different synthetic measurements**

| Synthetic measurement | Description | Snow cover fraction ($c_{snow}$) |
|---|---|---|
| snow_free | Ground surfaces without snow, randomly mixed with vegetation and soil | $c_{snow} = 0\%$ |
| snow_pure | Ground surfaces completely covered by snow | $c_{snow} = 100\%$ |
| snow_domi | Ground surfaces randomly mixed with vegetation, soil and snow, but snow is dominant in landcover | $c_{snow} > 75\%$ |
| snow_rand | Ground surfaces randomly mixed with vegetation, soil and snow, without limitation for snow cover fraction | $0\% \leq c_{snow} \leq 100\%$ |

These 4 sets of synthetic measurements are taken as the input for RemoTAP to conduct the retrieval with the baseline RemoTAP setup (without snow BRDF) and the extended RemoTAP setup (with snow BRDF). During validation, $\tau_{550}$, $\omega_{550}$ and $AE_{440-870}$ are chosen as the main performance indicators. In our experiment, a 5-band (490 nm, 565 nm, 670 nm, 865 nm, 1020 nm) retrieval is conducted and the validation results for different synthetic situations are shown in Figure 2.

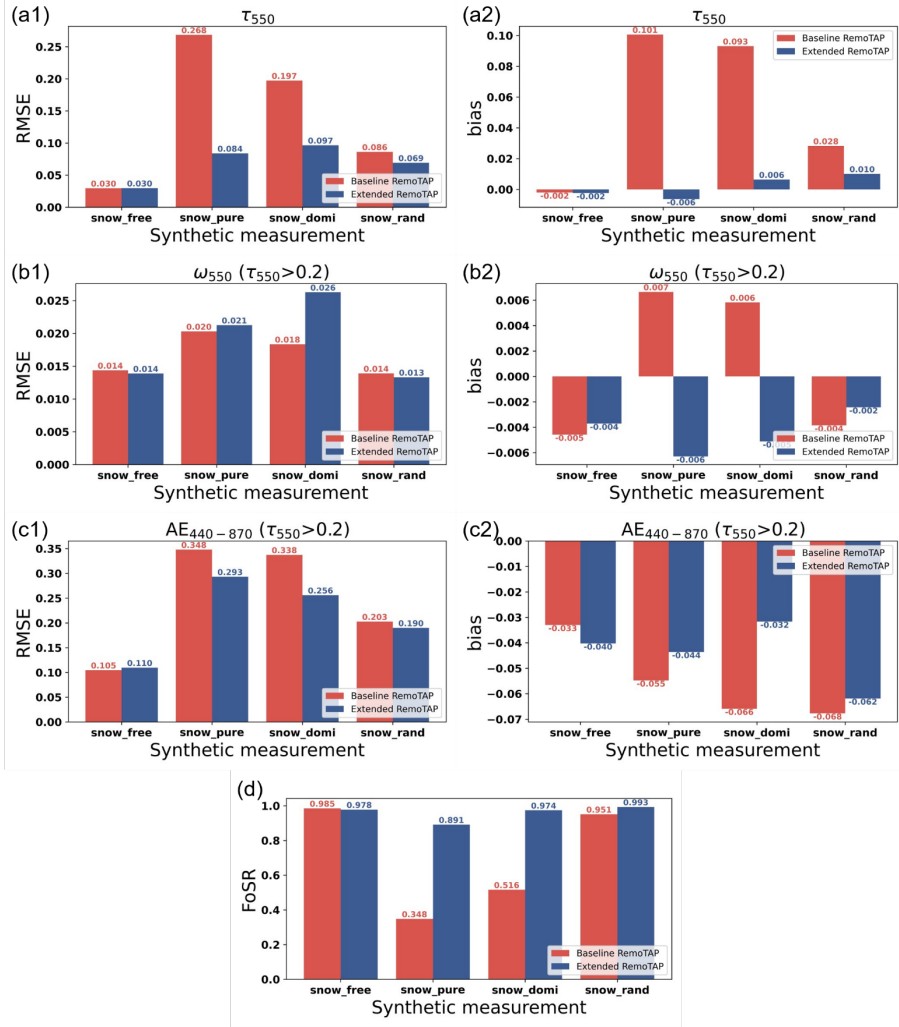

**Figure 2: Synthetic data retrievals of $\tau_{550}$, $\omega_{550}$ and $AE_{440-870}$ among baseline RemoTAP (red bar) and extended RemoTAP (blue bar) over different surfaces (see Table 4). Panels (a1, a2), (b1, b2), (c1, c2) and (d) show the bar-plots of $\tau_{550}$, $\omega_{550}$, $AE_{440-870}$ and fraction of successful retrievals (FoSR) respectively. Panels (a1, b1, c1) and (a2, b2, c2) show RMSE and bias, respectively.**

Comparing the RMSE, bias and fraction of successful retrievals between the baseline- and extended RemoTAP in Figure 2, we can conclude that the baseline RemoTAP, which utilizes the Ross-Li model to characterize the ground surfaces, has poor capability to retrieve aerosol properties over snow. For synthetic measurements of pure snow ("snow_pure") and dominated by snow ("snow_domi"), the fraction of successful retrieval is low (34.8% and 51.6%, respectively) and the retrieval accuracy for the successfully-retrieved pixels is low, with an RMSE of 0.268 and 0.197 respectively for $\tau_{550}$. The extended version of RemoTAP with the Ross-Li-Snow model on the other hand has much better performance with high fraction of successful retrievals (89.1% to 97.4%), with an RMSE on $\tau_{550}$ of 0.084 and 0.097 respectively. As expected, the performance over snow-covered pixels is still worse than for snow free pixels, because the high signal from the bright snow surface overwhelms the aerosol signal, leading to a reduced aerosol information content. For the synthetic measurements in "snow_rand", in which





snow is not dominant, the baseline RemoTAP gets acceptable performance, but the performance is still worse than the extended RemoTAP both in accuracy and fraction of successful retrievals. For $\omega_{550}$ retrieval, the performance of the baseline is slightly better than extended RemoTAP but it should be noted that only few pixels are left after the $\chi^2$ filtering. Overall, the results

demonstrate the importance of extending RemoTAP with the snow kernel in order to get the capability for aerosol retrieval over snow.

     Looking at the validation of the synthetic measurement "snow_free" in which there is no snow present, the extended RemoTAP maintains a similar capability as the baseline RemoTAP in terms of RMSE, bias and fraction of successful retrievals. For synthetic measurement with snow ("snow_pure", "snow_domi" and "snow_rand"), especially when snow is dominant

among the landcover ("snow_pure" and "snow_domi"), the extended RemoTAP has good performance for aerosol retrieval over snow. The scatter-plot for synthetic measurement "snow_pure" retrieved with extended RemoTAP is shown in Figure 3. The accuracy for $\tau_{550}$, $\omega_{550}$ and $AE_{440-870}$ retrievals is good, with an RMSE of 0.084 for $\tau_{550}$ retrieval, an RMSE of 0.021 for $\omega_{550}$ retrieval and an RMSE of 0.293 for $AE_{440-870}$ retrieval.

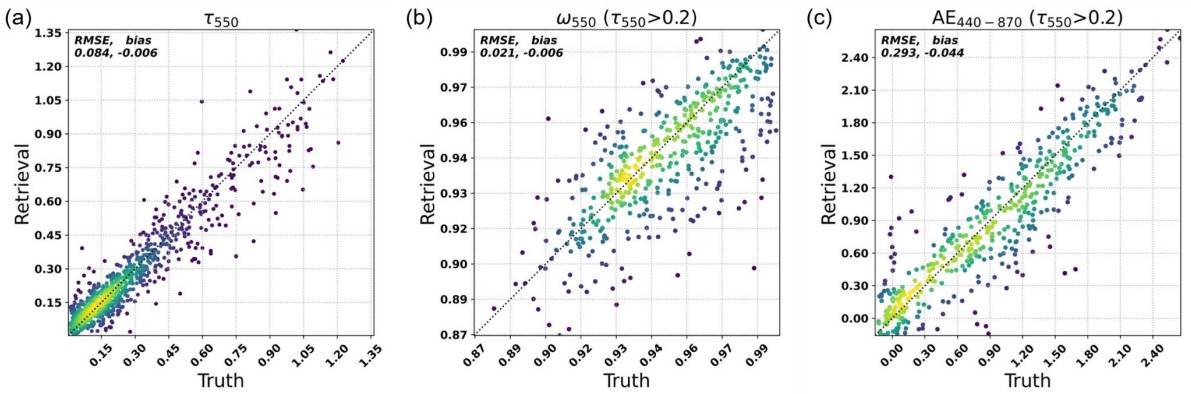

**Figure 3: Extended RemoTAP retrievals versus synthetic truth for $\tau_{550}$ (panel (a)), $\omega_{550}$ (panel (b)) and $AE_{440-870}$ (panel (c)) over pure snow surfaces ($c_{snow} = 100\%$)**

     The synthetic results suggest that PARASOL measurements have enough information to allow the inclusion of the snow kernel in the retrieval state vector. Also, they show that by inclusion of the snow kernel, the extended RemoTAP is capable of performing aerosol retrievals over snow in a consistent setup. The next section will show the performance of real measurements.

**5 Real data experiments**

In our experiments, 5-band (490 nm, 565 nm, 670 nm, 865 nm, 1020 nm), 4-band (490 nm, 565 nm, 670 nm, 865 nm) and 3-band (490 nm, 565 nm, 670 nm) retrievals are conducted with the extended RemoTAP, as well as with the baseline RemoTAP. The validation of $\tau_{550}$ for different retrieval setups over snow-dominant surfaces ($c_{snow} > 75\%$) is shown in Figure 4.

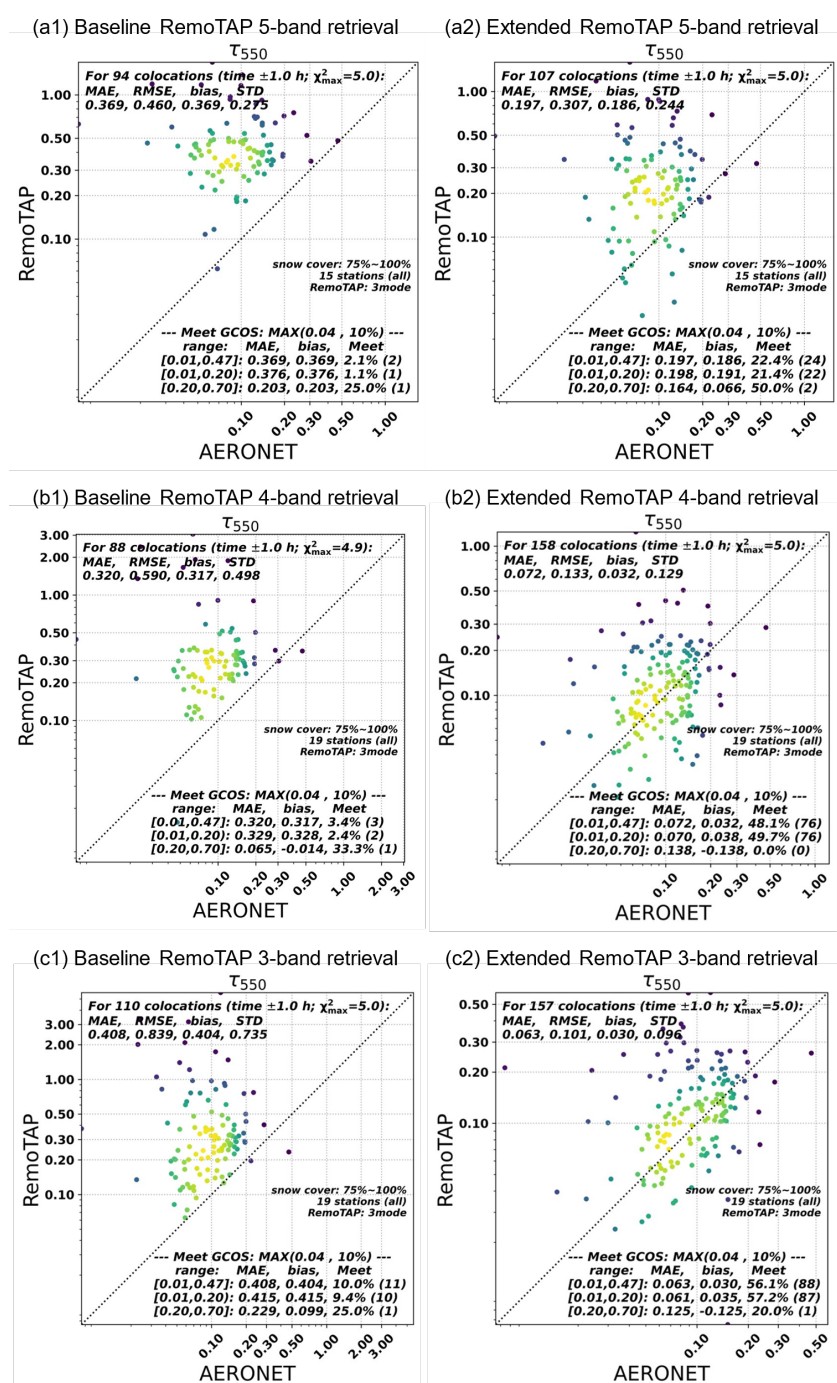

**Figure 4: Real data retrievals of $\tau_{550}$ among 5-band (panels (a1, a2)), 4-band (panels (b1, b2)) and 3-band (panels (c1, c2)) RemoTAP versus AERONET over snow-dominant surfaces ($c_{snow} > 75\%$). Panels (a1, b1, c1) show the scatter-plot of baseline RemoTAP and the panels (a2, b2, c2) show the scatter-plot of extended RemoTAP. The statistics with threshold required by Global Climate Observing System (GCOS) are also shown on the right bottom of each panel.**





According to the validation with AERONET for $\tau_{550}$ retrieval over snow-dominant surfaces shown in Figure 4, we find that:

(1) For the 5-band retrieval, the extended RemoTAP offers a better performance than the baseline RemoTAP, but the accuracy is still far from acceptable. This is caused by the fact that the retrieval algorithm has difficulty in fitting the 1020 nm band, where snow reflectance decreases significantly, making the reflectance much lower than that of the visible bands (see Figure 1). In order to get a better result than the 5-band retrieval, we investigate retrievals with a reduced number of spectral bands. First of all, the 1020 nm band is excluded because the algorithm has difficulties in fitting the strong spectral change

between the visible bands and the 1020 nm band (synthetic experiments indicate that this only works with a very accurate first guess value for the BRDF in the 1020nm band). We also investigate an even more reduced set of wavelength bands including only the 490 nm, 565 nm, and 670 nm bands (3-band retrieval). The performance of the 3 band and 4 band retrieval is much better than that of the 5-band retrieval, e.g., for the 4-band retrieval the RMSE is reduced from 0.307 to 0.133). Interestingly, for the 3-band retrieval the RMSE is further reduced to 0.101. We do not show the scatter-plot validation figures for SSA and

AE retrievals over snow-dominant surfaces ($c_{\text{snow}} > 75\%$) because not enough AERONET data are available for validation for snow-dominated pixels.

    (2) The extended RemoTAP has a significantly better agreement with AERONET than the baseline RemoTAP for the 3-, 4-, and 5-band retrievals. We would also like to emphasize that the fraction of pixels which meet the GCOS accuracy requirement also has a huge increase compared to the baseline RemoTAP of up to 45 percent-points for 3- and 4-band retrievals.

This demonstrates the importance of adding the snow kernel to our BRDF model for retrievals over snow.

    Figure 6 summarizes the RMSE and bias on $\tau_{550}$, $\omega_{550}$ and $AE_{440-870}$ for different retrieval setups for both snow-free ($c_{\text{snow}} = 0\%$) and snow-covered pixels ($0\% < c_{\text{snow}} \leq 100\%$). For SSA, we would like to note again that the statistics over snow have limited value and no strong conclusions should be drawn from these numbers. For AE, although there is not enough statistics over snow-dominant surfaces ($c_{\text{snow}} > 75\%$), there are enough pixels for validation when $0\% < c_{\text{snow}} \leq 100\%$.





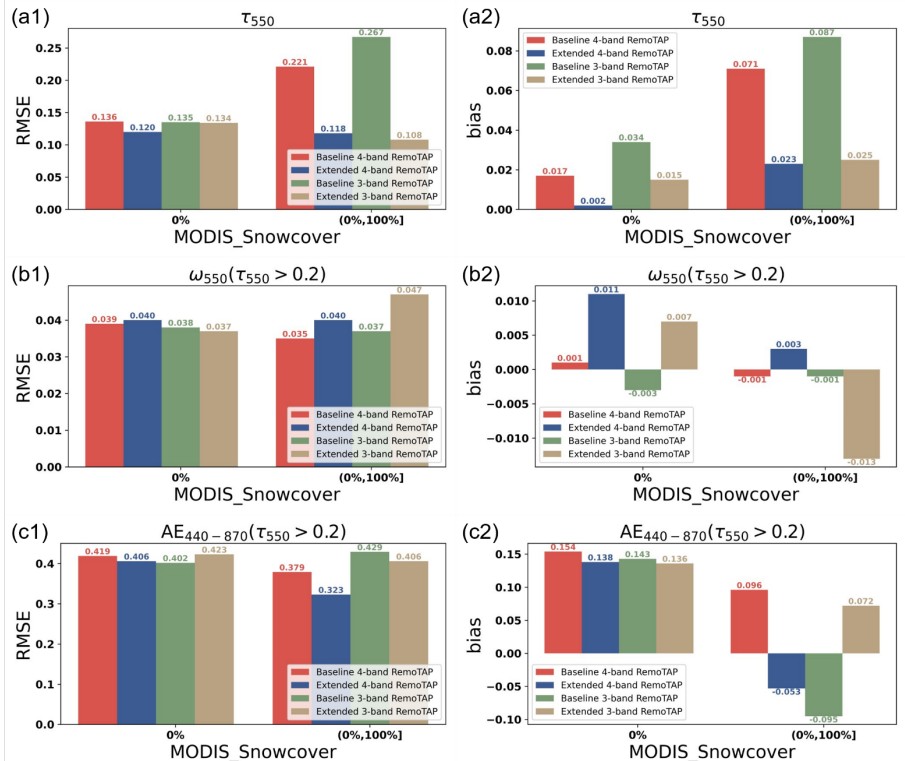


**Figure 5: Real data retrievals of $\tau_{550}$, $\omega_{550}$ and $AE_{440-870}$ for the baseline 4-band RemoTAP (red bar), extended 4-band RemoTAP (blue bar), baseline 3-band RemoTAP (green bar) and extended 3-band RemoTAP (brown bar) over snow surfaces ($0\% < c_{snow} \leq 100\%$) and snow-free surfaces ($c_{snow} = 0\%$). Panels (a1, a2), (b1, b2) and (c1, c2) show the bar-plots of $\tau_{550}$, $\omega_{550}$ and $AE_{440-870}$, respectively. Panels (a1, b1, c1) and (a2, b2, c2) show RMSE and bias, respectively.**


According to the validation results for retrievals over snow-free surfaces ($c_{snow} = 0\%$) and snow surfaces ($0\% < c_{snow} \leq 100\%$) shown in Figure 5, we can conclude that:

(1) For snow-free surfaces ($c_{snow} = 0\%$), the extended RemoTAP offers similar performance as the baseline RemoTAP for 3-band and 4-band retrievals, respectively. The only exception is the bias of $\omega_{550}$ retrieval, where extended RemoTAP
retrievals have larger bias than baseline RemoTAP, but the difference is small ($< 0.01$), especially given also the AERONET uncertainty on SSA and the limited number of validation points for SSA.

(2) For snow-covered surfaces ($0\% < c_{snow} \leq 100\%$), the baseline RemoTAP 4-band and 3-band retrievals fail to provide useful results with good accuracy, while both 3-band and 4-band extended RemoTAP retrievals are able to have good performance.

(3) Comparing the extended RemoTAP for 3- and 4-band retrievals, the results of $\tau_{550}$ retrievals are similar, but the 4-band extended RemoTAP retrieval has slightly better performance for $\omega_{550}$ and $AE_{440-870}$ retrievals (comparing the blue and brown bars). Therefore, the 4-band extended RemoTAP retrieval is recommended for aerosol retrieval over snow from PARASOL.

In addition to the advantage in accuracy, taking 4-band RemoTAP as an example, there is also an obvious increase in the fraction of successful retrievals (chi2 < 5), as shown in Fig. 6 for different snow fractions bins. On average, inclusion of the

snow kernel leads to an increase in successful retrievals by 5.8 percent-points and when the snow cover is large this increases to 18.9 percent-points, compared to the baseline, resulting in a factor ~2 more successful retrievals. This indicates that the extended RemoTAP provides better performance on both accuracy and goodness-of-fit for aerosol retrieval over snow.

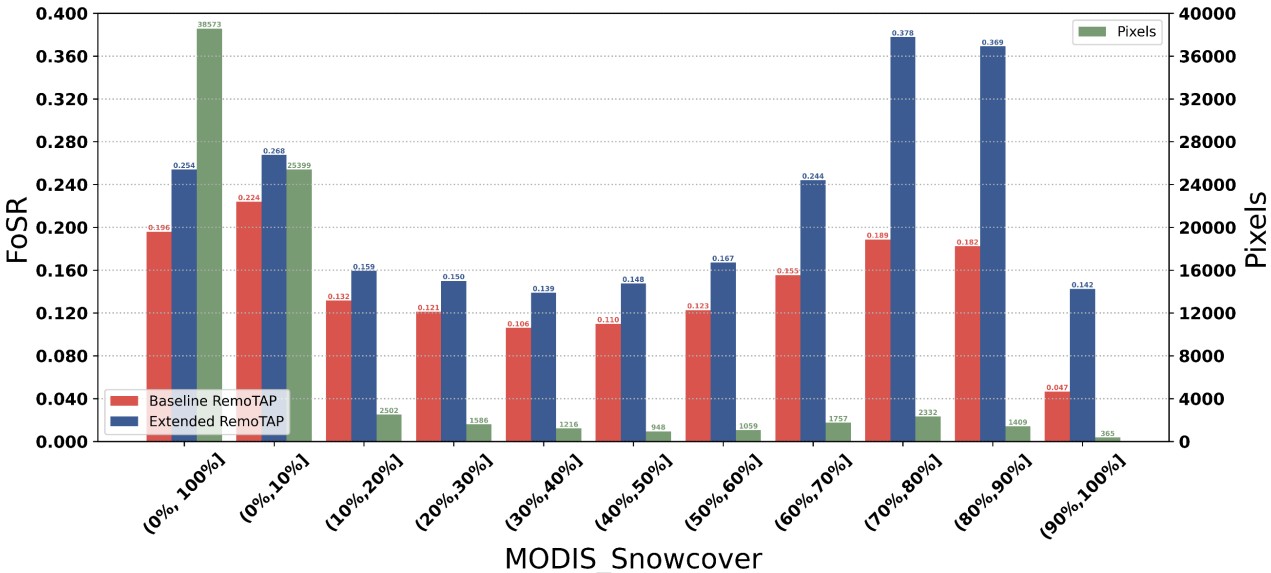

**Figure 6: Fraction of successful retrievals (left y-axis) for real data retrievals among baseline 4-band RemoTAP (red bar) and**
**extended 4-band RemoTAP (blue bar) over different snow cover intervals. The number of the matched snow pixels (right y-axis) over different snow cover intervals is plotted in green bar.**

Figure 7 shows an example of the difference between PARASOL measurements and the RemoTAP forward model simulations (after convergence) for ToA reflectance and DoLP for aerosol retrieval over a snow-dominated pixel. Comparing

the ToA reflectance for the baseline- and extended RemoTAP in Figure 7 (a1) and (b1), the difference between PARASOL measurements and the results of extended RemoTAP forward model is much smaller than that of baseline RemoTAP. Comparing the ToA DoLP in Figure 7 (a2) and (b2), the performance is quite comparable, and there is slight advantage at the 865 nm band for the extended RemoTAP. It is important to note that there are some unphysical oscillations in the PARASOL measurements which are probably caused by interpolation error in the PARASOL level 1C processing for inhomogeneous

scenes.

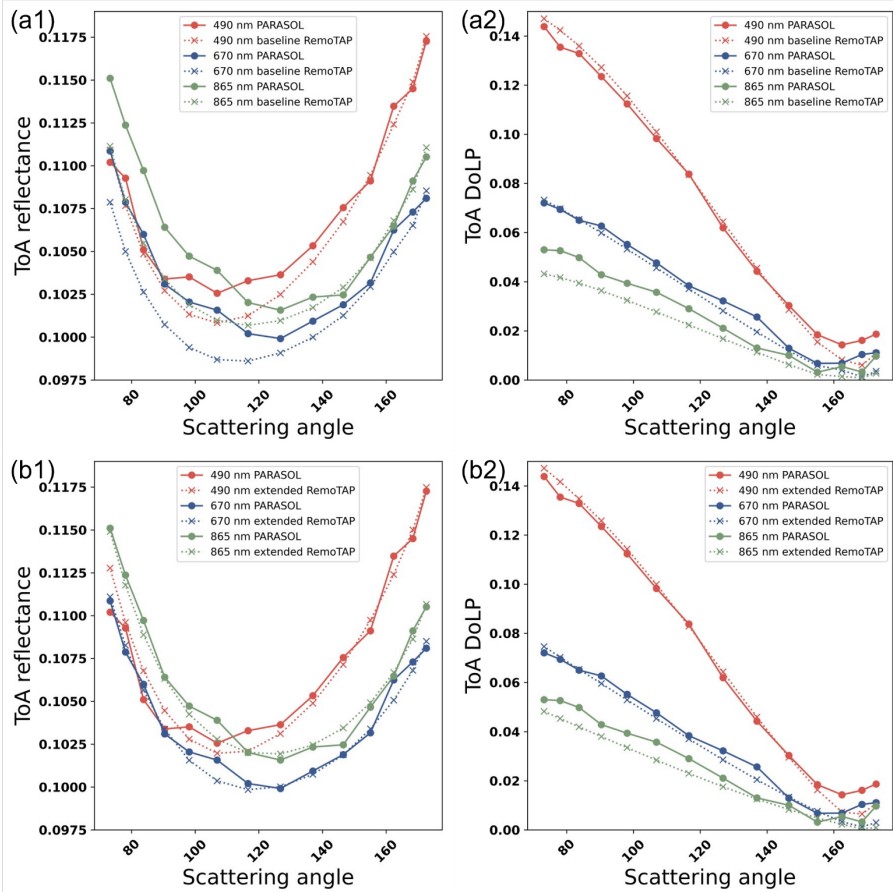

**Figure 7: Example of PARASOL measurements and the RemoTAP results for ToA reflectance (panel (a1, b1)) and ToA DoLP (panel (a2, b2)). Solid lines in panels (a1, a2, b1, b2) refer to PARASOL measurements, dotted lines in panel (a1, a2) refer to baseline RemoTAP result and dotted lines in panel (b1, b2) refer to extended RemoTAP results. The red lines refer to 490 nm band, the blue lines refer to 670 nm band and the green lines refer to 865 nm band. The pixel is located at (43.75° N, 96.65° W), and the snow cover of this pixel is 68.41%.**

## 6 Conclusions

We extended the RemoTAP algorithm with a snow kernel in the BRDF model to carry out aerosol retrieval over snow from PARASOL MAP measurements. We conducted synthetic retrievals to investigate the necessity and advantage of extending RemoTAP, and conducted retrievals on real measurements. For PARASOL retrievals with 4-bands (490 nm, 565 nm, 670 nm, 865 nm), the extended RemoTAP retrieval results agree well with AERONET of optical properties, both for retrievals over snow as well as for snow-free areas. The RMSE, bias, and fraction of retrievals within the GCOS requirements are 0.118, 0.023, and 57.2% respectively for $\tau_{550}$ retrieval over partly snow-covered surfaces. This is much improved compared to the baseline RemoTAP (without snow kernel) that yields an RMSE, bias, and fraction of retrievals within the GCOS requirements of 0.221, 0.071, and 48.2% respectively. Furthermore, the fraction of successful retrievals also improves by up to 18.9 percent-



points compared with the baseline. The improvement is most striking for surfaces that have snow cover > 75%, where the number of successful retrievals more than doubles and the fraction of retrievals that are within the GCOS requirement increases from 3.4% to 48.1%. The performance of the extended RemoTAP on accuracy and goodness-of-fit is in good agreement with the expectation from the synthetic data experiments. A limitation of the extended RemoTAP is that it is not able to fit

PARASOL measurements at 1020 nm, where the snow albedo is substantially lower than at lower wavelengths. Therefore, the 4-band extended RemoTAP is recommended as the best choice for aerosol retrieval over snow surfaces considering the performance for AOT, SSA and AE retrieval, both from synthetic- and real retrievals.

*Data availability.* The PARASOL level-1 data can be downloaded from the website http://www.icare.univ-lille1.fr/parasol/products (last access: 8 June 2023). The AERONET data can be downloaded from the website https://aeronet.gsfc.nasa.gov/ (last access: 8 June 2023). The MERRA-2 meteorological data can be accessed through the website https://gmao.gsfc.nasa.gov/reanalysis/MERRA-2/ (last access: 8 June 2023). The MODIS/Aqua snow cover data can be accessed through the website https://nsidc.org/data/myd10a1/versions/61 (last access: 8 June 2023).


*Author contributions.* ZZ, GF and OH designed the experiments. ZZ performed the experiments and data analysis and GF and OH contributed to the interpretation. GF and OH implemented the snow kernel in RemoTAP. ZZ wrote the first paper draft which was finalized by contributions from GF and OH.

*Competing interests.* At least one of the (co-)authors is a member of the editorial board of Atmospheric Measurement Techniques.

*Acknowledgements.* Zihan Zhang is supported by a scholarship from China Scholarship Council (CSC).

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
