# Peer review of "Aerosol retrieval over snow using RemoTAP"

_Atmospheric Measurement Techniques, 2023_

## Referee Comment (RC2)

An excellent structured work, proving the necessity of introducing the BRDF in the RemoTAP algorithm for accurate aerosol retrievals over snow surfaces. The newly developed extended RemoTAP algorithm is tested against both synthetic and real PARASOL retrievals showing promising performance.

**Line 83:** Snow kernel function is given below the equations (6-7). I would say it is better to refer it and show its equation only on $87^{th}$ line as it is.

**Line 68:** It's unclear to me if with the term $z_{aer}$ you mean the top or the bottom of the aerosol layer. Other retrievals separate these two terms so here it's a bit confusing.

**Table 1:** For the $3^{rd}$ Mode add the fixed $z_{aer}$ as it is noted on line 74, so the reader doesn't have to search for this information in the text.

**3.5 Data pre-processing**

AERONET utilized for collocation purposes is introduced in the previous paragraph, but I think it would be better if you add AND here something like "in terms of AOT, SSA, AE" because I had to look up to the AERONET data to remember this.

**Table 3:** The minimum and maximum values give the range of the randomly generated input parameters? Is there a reason you choose these limits?

**Figure 2:** The general better performance of extended RemoTAP is clear. If you changed the $\chi 2$ limit for the filtering do you think it would have better results for SSA (maybe a sensitivity test)? The difference of 0.008 (RMSE) between baseline and extended RemoTAP over snow_domi surfaces appears small but in terms of % relative difference it is not insignificant. I'm thinking if it would be more proper using the baseline RemoTAP for SSA.

Figure 3: Why x-axis is labeled as Truth and not AERONET and y-axis as Retrieval and not Extended RemoTAP? I had to read the caption to understand the figure.

---

## Author Comment (AC1)

**Reply to Referee #1:**

We thank the referee for the constructive and positive review.

1. *Line 58: Make it clear the e is not only the measurement error, but also contain the modeling error.*

   **Response:**

   We agree. We have revised it in '2.1 Forward model':

   'where $\boldsymbol{e_y}$ refers to the error vector including measurement error and modelling error.'

2. *Line 63: I understand that the RemoTAP algorithm uses the degree of polarization to quantify his piece of information. In my opinion, it would have been better to use the Stokes parameter Q (normalized similarly to I) with the plane of scattering as a reference. Indeed, Q then contains both the intensity of the polarized reflectance and some information about the direction (perpendicular versus parallel to the scattering plane).*

   **Response:**

   We believe both approaches have advantages and disadvantages. The advantage of using *DoLP* is that it is a relative quantity where multiplicative (calibration) errors in *I*, *Q*, and *U* cancel out. The disadvantage is to have 2 quantities in the measurement vector (*I*, *DoLP*) that are not independent. Using *Q* in the scattering plane (or polarized radiance) is more sensitive to calibration errors but has the advantage of having independent quantities in the measurement vector. In the past we have performed some sensitivity studies for PARASOL where using *DoLP* had a slight advantage for aerosol retrievals.

3. *Table 1: I am a bit surprised to see so many aerosol parameters that are retrieved. In particular, I doubt that there is sufficient information in the data to estimate the aerosol height for three different modes. Is there any value in the retrieved aerosol height properties? Also, there is no height for mode 3, opposite to mode 1 and 2. Any reason for that?*

   **Response:**

   We agree that PARASOL data provide weak feasibility to retrieve ALH because the shortest wavelength polarization band (490 nm) is quite far from the UV. However, there is still some sensitivity, and we see the retrieval of other aerosol properties improves of we include ALH in the state vector. Mode 3 represents sea salt which we assume is in the boundary layer and we do not retrieve ALH. Mode 1 (fine mode) and mode 2 (representative for Dust) can consist of elevated layers. Although indeed we do not have independent information on ALH for mode 1 and 2 separately, it still has advantage to include them separately in the state vector, for cases where one of the 2 modes dominates the aerosol distribution.

4. *I understand that the modeling assumes that the surface reflectance BRDF shape does not vary with wavelength. This a strong assumption. Indeed, the surface reflectance amplitude varies strongly with wavelength and high albedos tend to generate more isotropic reflectance than low albedos. Would it be possible to add some freedom in the spectral variation of the BRDF modeling?*

   **Response:**

In our algorithm, the wavelength-dependence for the isotropic reflectance $A(\lambda)$ has been already considered. The isotropic reflectance $A(\lambda)$ is fitted for different bands, and it is shown in equation (5) in '2.1 Forward model'. However, the kernel coefficients ($k_{geo}$, $k_{vol}$, $k_{snow}$ in Eq. 5) do not depend on wavelength in RemoTAP and it would be a major code change to modify that. Based on Litvinov et al. (2011) this assumption is justified for vegetation and soil surfaces. Given that we can fit the PARASOL measurements well between 444-865 nm this assumption also does not seem to be a limitation for snow surfaces in the VNIR. The problem with the 1020 nm band in our retrievals is not caused by the assumption of a constant $k_{snow}$ because we can reproduce it with consistent synthetic measurements.

5. *Line 165: In the real world, the measurement of DoLP is more noisy in case of low reflectances than with high reflectances. It is then unfortunate to use a fixed DoLP uncertainty that does not depend on the scene*

   We agree this is a limitation of this setup which and we will investigate improvements in the near future. On the other hand, in the inversion approach of RemoTAP it is the balance between noise in radiance and DoLP that is of importance and not so much the assumed value itself.

6. *Table 3: What are the rationale for setting these min and max? Why some values are missing?*

   **Response:**

   The min and max values are based on an inspection of global PARASOL retrievals, where appropriate. We revised the table and the following paragraph to give more details and avoid misunderstanding:

   '

**Table 3: Observation geometry, aerosol properties and surface properties used to create synthetic PARASOL observations. $c_{veg}$, $c_{soil}$ and $c_{snow}$ are the fraction of vegetation, soil and snow respectively. Distribution 'linear' refers to $X \sim U(X_{min}, X_{max})$, and distribution 'logarithmic' refers to $\ln X \sim U(\ln X_{min}, \ln X_{max})$, where $X$ is the property value, $X_{min}$ and $X_{max}$ are the minimum and maximum respectively.**

| Property | Minimum | Maximum | Distribution |
|---|---|---|---|
| $\theta_s$ | 10 | 70 | logarithmic |
| $\theta_v$ | -65 | 65 | $0, \pm 10, \pm 20, \pm 30, \pm 40, \pm 50, \pm 60, \pm 65$ |
| $\varphi$ | 20 | 160 | $\varphi = 20$ when $\theta_v \geq 0$, $\varphi = 160$ when $\theta_v < 0$ |
| $c_{veg}$ | 0.0 | 1.0 | linear or fixed (see Table 4) |
| $c_{soil}$ | 0.0 | 1.0 | linear or fixed (see Table 4) |
| $c_{snow}$ | 0.0 | 1.0 | linear or fixed (see Table 4) |
| $\tau_{550}$ (mode 1) | 0.005 | 1.0 | logarithmic |
| $\tau_{550}$ (mode 2) | 0.0025 | 0.25 | logarithmic |
| $\tau_{550}$ (mode 3) | 0.0025 | 0.25 | logarithmic |
| $r_{eff}$ (mode 1) | 0.1 | 0.3 | linear |
| $r_{eff}$ (mode 2) | 0.8 | 1.5 | linear |
| $r_{eff}$ (mode 3) | 1.5 | 4.0 | linear |
| $v_{eff}$ (mode 1) | 0.1 | 0.3 | linear |
| $v_{eff}$ (mode 2) | 0.6 | 0.6 | fixed |
| $v_{eff}$ (mode 3) | 0.6 | 0.6 | fixed |
| $f_{sph}$ (mode 1) | 1.0 | 1.0 | fixed |

| | | | |
|---|---|---|---|
| $f_{sph}$ (mode 2) | 0.0 | 0.0 | fixed |
| $f_{sph}$ (mode 3) | 1.0 | 1.0 | fixed |
| $z_{aer}$ (mode 1) | 1000 | 6000 | linear |
| $z_{aer}$ (mode 2) | 1000 | 6000 | linear |
| $z_{aer}$ (mode 3) | 500 | 500 | fixed |

The surface properties in the synthetic data set are created by mixing the contribution of the surface reflection by vegetation, soil, and snow. By controlling the fraction of vegetation, soil and snow, 4 sets of synthetic measurements are created, where the detailed information for these 4 synthetic measurements are listed in Table 4. The isotropic reflectance $A(\lambda)$ is calculated with equation $A(\lambda) = c_{veg}A_{veg}(\lambda) + c_{soil}A_{soil}(\lambda) + c_{snow}A_{snow}(\lambda)$. $A_{veg}(\lambda)$. $A_{soil}(\lambda)$ and $A_{snow}(\lambda)$ refer to the reference reflectance spectra for vegetation, soil and snow (shown in Figure 1). For the kernel coefficients of Li-Sparse ($k_{geo} = 0.087c_{veg} + 0.158c_{soil}$) and Ross-Thick ($k_{vol} = 0.688c_{veg} + 0.547c_{soil}$), the constant values we use are found by Litvinov et al. (2011).'

7. *Figure 3 and 4: Explain color coding*

   **Response:**

   The explanation of color coding is added:

   'The color indicates the density of data points, where yellow indicates high density and blue/purple low density (viridis color map).'

**Reference:**

Litvinov, P., Hasekamp, O., and Cairns, B.: Models for surface reflection of radiance and polarized radiance: Comparison with airborne multi-angle photopolarimetric measurements and implications for modeling top-of-atmosphere measurements, Remote Sensing of Environment, 115, 781-792, https://doi.org/10.1016/j.rse.2010.11.005, 2011.

---

## Author Comment (AC2)

**Reply to Referee #2:**

We thank the referee for the constructive and positive review.

1.  *Line 83: Snow kernel function is given below the equations (6-7). I would say it is better to refer it and show its equation only on 87th line as it is.*

    **Response:**

    We agree that the equation (8) is the primary function for snow kernel but also Eq. 9 and 10 are important to include for the reader to know exactly how we used the kernel of Jiao et al. (2019). We removed the phrase 'in Eq. (8)' just before Eq (8) because it may suggest that it refers to Eq (8) of Jiao et al. (2019).

2.  *Line 68: It's unclear to me if with the term z_aer you mean the top or the bottom of the aerosol layer. Other retrievals separate these two terms so here it's a bit confusing.*

    **Response:**

    The term aerosol layer height (ALH) z_aer refers to the altitude of the centre of aerosol layer, that is z_aer = 0.5*(z_bottom+z_top). We add a definition at the place where ALH first appears in our manuscript:

    'aerosol layer height ($z_{aer}$, here refers to the altitude of the aerosol layer centre)'

3.  *Table 1: For the 3rd Mode add the fixed z_aer as it is noted on line 74, so the reader doesn't have to search for this information in the text.*

    **Response:**

    Thanks for your suggestion. We have added it.

4.  *AERONET utilized for collocation purposes is introduced in the previous paragraph, but I think it would be better if you add AND here something like "in terms of AOT, SSA, AE" because I had to look up to the AERONET data to remember this.*

    **Response:**

    Thanks for your suggestion. In this way, the readers do not have to look back and check for the information. We have revised it in '3.5 Data pre-processing':

    'The first step is to match global PARASOL L1 measurement data with global AERONET validation data (AOT, SSA & AE).'

5.  *Table 3: The minimum and maximum values give the range of the randomly generated input parameters? Is there a reason you choose these limits?*

    **Response:**

    The min and max values are based on an inspection of global PARASOL retrievals, where appropriate. We revised the table and the following paragraph to give more details and avoid misunderstanding:

    '

**Table 3: Observation geometry, aerosol properties and surface properties used to create synthetic PARASOL observations. $c_{veg}$, $c_{soil}$ and $c_{snow}$ are the fraction of vegetation, soil and snow respectively. Distribution 'linear' refers to $X \sim U(X_{min}, X_{max})$, and distribution 'logarithmic' refers to $\ln X \sim U(\ln X_{min}, \ln X_{max})$, where $X$ is the property value, $X_{min}$ and $X_{max}$ are the minimum and maximum respectively.**

| Property | Minimum | Maximum | Distribution |
|---|---|---|---|
| $\theta_s$ | 10 | 70 | logarithmic |
| $\theta_v$ | -65 | 65 | $0, \pm 10, \pm 20, \pm 30, \pm 40, \pm 50, \pm 60, \pm 65$ |
| $\varphi$ | 20 | 160 | $\varphi = 20$ when $\theta_v \geq 0$, $\varphi = 160$ when $\theta_v < 0$ |
| $c_{veg}$ | 0.0 | 1.0 | linear or fixed (see Table 4) |
| $c_{soil}$ | 0.0 | 1.0 | linear or fixed (see Table 4) |
| $c_{snow}$ | 0.0 | 1.0 | linear or fixed (see Table 4) |
| $\tau_{550}$ (mode 1) | 0.005 | 1.0 | logarithmic |
| $\tau_{550}$ (mode 2) | 0.0025 | 0.25 | logarithmic |
| $\tau_{550}$ (mode 3) | 0.0025 | 0.25 | logarithmic |
| $r_{eff}$ (mode 1) | 0.1 | 0.3 | linear |
| $r_{eff}$ (mode 2) | 0.8 | 1.5 | linear |
| $r_{eff}$ (mode 3) | 1.5 | 4.0 | linear |
| $v_{eff}$ (mode 1) | 0.1 | 0.3 | linear |
| $v_{eff}$ (mode 2) | 0.6 | 0.6 | fixed |
| $v_{eff}$ (mode 3) | 0.6 | 0.6 | fixed |
| $f_{sph}$ (mode 1) | 1.0 | 1.0 | fixed |
| $f_{sph}$ (mode 2) | 0.0 | 0.0 | fixed |
| $f_{sph}$ (mode 3) | 1.0 | 1.0 | fixed |
| $z_{aer}$ (mode 1) | 1000 | 6000 | linear |
| $z_{aer}$ (mode 2) | 1000 | 6000 | linear |
| $z_{aer}$ (mode 3) | 500 | 500 | fixed |

The surface properties in the synthetic data set are created by mixing the contribution of the surface reflection by vegetation, soil, and snow. By controlling the fraction of vegetation, soil and snow, 4 sets of synthetic measurements are created, where the detailed information for these 4 synthetic measurements are listed in Table 4. The isotropic reflectance $A(\lambda)$ is calculated with equation $A(\lambda) = c_{veg}A_{veg}(\lambda) + c_{soil}A_{soil}(\lambda) + c_{snow}A_{snow}(\lambda)$. $A_{veg}(\lambda)$. $A_{soil}(\lambda)$ and $A_{snow}(\lambda)$ refer to the reference reflectance spectra for vegetation, soil and snow (shown in Figure 1). For the kernel coefficients of Li-Sparse ($k_{geo} = 0.087c_{veg} + 0.158c_{soil}$) and Ross-Thick ($k_{vol} = 0.688c_{veg} + 0.547c_{soil}$), the constant values we use are found by Litvinov et al. (2011).'

6. *Figure 2: The general better performance of extended RemoTAP is clear. If you changed the $\chi 2$ limit for the filtering do you think it would have better results for SSA (maybe a sensitivity test)? The difference of 0.008 (RMSE) between baseline and extended RemoTAP over snow_domi surfaces appears small but in terms of % relative difference it is not insignificant. I'm thinking if it would be more proper using the baseline RemoTAP for SSA.*

**Response:**

Here are two reasons why we recommend 4-band extended RemoTAP for aerosol retrieval over snow: (1) RemoTAP is a full-physical algorithm, so AOT and SSA are not retrieved separately. We first retrieve microphysical properties of aerosol and then calculate and output AOT and SSA. Therefore, using two versions of RemoTAP to retrieve AOT and SSA harms the microphysical relationship between AOT and SSA and would further lead to the contradictions related with physical basis; (2) RemoTAP algorithm will be used to generate the global aerosol products of

NASA SPEXone/PACE, so we have to consider the processing speed. Since the global data processing is time-consuming, if we output SSA and AOT product with two versions of RemoTAP, it would double the processing time, therefore we would prefer to choose one comprehensively good-performance version of RemoTAP to generate both SSA and AOT products in the same run.

7. *Figure 3: Why x-axis is labeled as Truth and not AERONET and y-axis as Retrieval and not Extended RemoTAP? I had to read the caption to understand the figure.*

**Response:**

Figure 3 belongs to the section '4 Synthetic data experiments', so the x-axis does not refer to AERONET data but refer to the truth which is used to simulate the synthetic measurements. In addition, Figure 4 belongs to the section '5 Real data experiments' and the x-axis refers to AERONET data. In order to better distinguish these two figures and avoid misleading, we revised Figure 3 with new axis labels and figure caption:

'

[Figure]

**Figure 3: Synthetic data retrievals of $\tau_{550}$, $\omega_{550}$ and $AE_{440-870}$ among extended RemoTAP retrievals versus synthetic truth over pure snow surfaces ($c_{snow} = 100\%$). Panels (a, b, c) show the scatter-plot of $\tau_{550}$, $\omega_{550}$ and $AE_{440-870}$, respectively.**

'

**Reference:**

Jiao, Z., Ding, A., Kokhanovsky, A., Schaaf, C., Bréon, F.-M., Dong, Y., Wang, Z., Liu, Y., Zhang, X., Yin, S., Cui, L., Mei, L., and Chang, Y.: Development of a snow kernel to better model the anisotropic reflectance of pure snow in a kernel-driven BRDF model framework, Remote Sensing of Environment, 221, 198-209, https://doi.org/10.1016/j.rse.2018.11.001, 2019.

Litvinov, P., Hasekamp, O., and Cairns, B.: Models for surface reflection of radiance and polarized radiance: Comparison with airborne multi-angle photopolarimetric measurements and implications for modeling top-of-atmosphere measurements, Remote Sensing of Environment, 115, 781-792, https://doi.org/10.1016/j.rse.2010.11.005, 2011.

---

## Author Response (AR2)

**Reply to Associate Editor:**

We thank the associate editor for the constructive and positive review.

1. *Your choice of the snow spectrum A(lambda) is not clear. From my point of view, the best choice is given by Eq. (2) in the paper listed below. The parameter L in Eq. (2) can be included in the retrieval parameters list. For the case of polluted snow, the modification is needed (A. Kokhanovsky, Snow Optics book).*
   *Kokhanovsky, A.A.; Brell, M.; Segl, K.; Bianchini, G.; Lanconelli, C.; Lupi, A.; Petkov, B.; Picard, G.; Arnaud, L.; Stone, R.S.; et al. First Retrievals of Surface and Atmospheric Properties Using EnMAP Measurements over Antarctica. Remote Sens. 2023, 15, 3042.*
   *https://doi.org/10.3390/rs15123042*

   **Response:**

   The formula $A(\lambda) = c_{veg}A_{veg}(\lambda) + c_{soil}A_{soil}(\lambda) + c_{snow}A_{snow}(\lambda)$ is only used to generate synthetic measurements based on land cover fraction. When generating the synthetic truth, the choice of $A_{soil}(\lambda)$, $A_{veg}(\lambda)$ and $A_{snow}(\lambda)$ is given in detail at the caption of Figure 1, and these 3 spectra are all based on real measurements for vegetation, soil, and snow. We believe that these spectra are sufficient for use given the limited scope of the synthetic study.

   'Figure 1: Reference reflectance for vegetation, soil and snow. The snow data are downloaded from National Snow and Ice Data Center (NSIDC) (https://nsidc.org/data/hma_sbrf/versions/1) and the soil and vegetation data are downloaded from ASTER spectral library (https://speclib.jpl.nasa.gov/).'

   When we conduct the retrieval with real measurement data, the $A(\lambda)$ refers to the isotropic reflectance for the land cover, which includes the contribution of all the land cover types, and instead of giving a formula of $A(\lambda)$ and fit the formula parameters, we fit $A(\lambda)$ directly for each wavelength separately which gives full flexibility to represent any spectral shape of the snow albedo. In the revised version, we include a reference to Kokhanovsky et al. (2023) to discuss an alternative method of retrieval where the spectral dependence of $A(\lambda)$ is parameterized. We have include it in '2.1 Forward model':

   'In our algorithm, $A(\lambda)$ is fit separately for each wavelength which provides full flexibility to represent any spectral shape of the snow albedo. An alternative method to deal with the spectral dependence is discussed by Kokhanovsky et al. (2023) where $A(\lambda)$ is parameterized with effective absorption length L which is valid for snow with different microstructure and pollution level.'

2. *To my understanding your snow spectrum is valid for a given snow microstructure (parameter L in Eq.2 in the paper given below) and some level of pollution. Indeed, the decrease of reflectance towards shorter wavelengths is a clear indication of the fact that snow is loaded by dust. Actually, the dust in air will produce the same effect (decrease of reflectance towards UV). How do you distinguish these two cases (dust in snow and dust in air) in your retrieval technique?*

   **Response:**

   We agree for the case of single-view radiometric remote sensing. However, since we are using multi-angle measurements of radiance and polarization, our retrieval has better capability to distinguish dust in snow and dust in the air. For example, through interaction with Rayleigh scattering, absorption by aerosols in the atmosphere has a characteristic effect on polarization measurements near 90° scattering angle. In our synthetic study we did not encounter specific difficulties for cases with strong dust loading.

**Reference:**

Kokhanovsky, A. A., Brell, M., Segl, K., Bianchini, G., Lanconelli, C., Lupi, A., Petkov, B., Picard, G., Arnaud, L., Stone, R. S., and Chabrillat, S.: First Retrievals of Surface and Atmospheric Properties Using EnMAP Measurements over Antarctica, Remote Sensing, 15, 3042, 2023.